# Parasite Load and Site-Specific Parasite Pressure as Determinants of Immune Indices in Two Sympatric Rodent Species

**DOI:** 10.3390/ani9121015

**Published:** 2019-11-22

**Authors:** Tim R. Hofmeester, Esther J. Bügel, Bob Hendrikx, Miriam Maas, Frits F. J. Franssen, Hein Sprong, Kevin D. Matson

**Affiliations:** 1Resource Ecology Group, Wageningen University, Droevendaalsesteeg 3a, 6708PB Wageningen, The Netherlands; tim.hofmeester@slu.se (T.R.H.); e.j.bugel@iclon.leidenuniv.nl (E.J.B.); b.hendrikx91@gmail.com (B.H.); 2Centre for Infectious Disease Control, National Institute for Public Health and the Environment, Antonie van Leeuwenhoeklaan 9, 3721 MA Bilthoven, The Netherlands; miriam.maas@rivm.nl (M.M.); frits.franssen@rivm.nl (F.F.J.F.); hein.sprong@rivm.nl (H.S.)

**Keywords:** rodents, ecological immunology, natural antibodies, haptoglobin, neutrophil to lymphocyte ratio, immune strategy, vector-borne pathogens, parasitology, zoonosis

## Abstract

**Simple Summary:**

Wild animals can host diseases that affect humans (i.e., zoonotic diseases). However, not all wild animals are equal in their hosting capacities. In fact, the immune system, the main defense against diseases, varies within and among species. Within species, variation relates to two factors: parasite load and parasite pressure. Parasite load refers to the amount of parasites in or on an individual. Parasite pressure refers to the amount of parasites in a location. The importance of these factors in shaping the immune system of wild rodents, a group of animals known to host zoonotic diseases, is poorly understood. Overall, the rodent species we studied (bank voles and wood mice) hosted 5 microparasites, 9 ectoparasites, and 8 gastrointestinal parasites. We found that parasite load and parasite pressure related to different facets of the immune system. We also found that bank voles exhibited higher levels of one immune defense than wood mice, but higher levels of this defense correlated with a worm infection only in wood mice. Lastly, a white blood cell ratio correlated with infection by a zoonotic parasite. Studies like ours help to document the complexities of host–parasite interactions and how these interactions shape zoonotic disease risk in a changing world.

**Abstract:**

Wildlife is exposed to parasites from the environment. This parasite pressure, which differs among areas, likely shapes the immunological strategies of animals. Individuals differ in the number of parasites they encounter and host, and this parasite load also influences the immune system. The relative impact of parasite pressure vs. parasite load on different host species, particularly those implicated as important reservoirs of zoonotic pathogens, is poorly understood. We captured bank voles (*Myodes glareolus*) and wood mice (*Apodemus sylvaticus*) at four sites in the Netherlands. We sampled sub-adult males to quantify their immune function, infestation load for ecto- and gastrointestinal parasites, and infection status for vector-borne microparasites. We then used regression trees to test if variation in immune indices could be explained by among-site differences (parasite pressure), among-individual differences in infestation intensity and infection status (parasite load), or other intrinsic factors. Regression trees revealed splits among sites for haptoglobin, hemagglutination, and body-mass corrected spleen size. We also found splits based on infection/infestation for haptoglobin, hemolysis, and neutrophil to lymphocyte ratio. Furthermore, we found a split between species for hemolysis and splits based on body mass for haptoglobin, hemagglutination, hematocrit, and body-mass corrected spleen size. Our results suggest that both parasite pressure and parasite load influence the immune system of wild rodents. Additional studies linking disease ecology and ecological immunology are needed to understand better the complexities of host–parasite interactions and how these interactions shape zoonotic disease risk.

## 1. Introduction

Free-living wild animals are repeatedly exposed to different parasites. These parasites can be found on plants and other animals, in soil and water, and generally throughout the animal’s environment [1,2]. When an animal interacts with one or more parasites, that animal’s immune system can respond in different ways [3]. For example, some hosts engage in strategies of resistance, while others rely on tolerance [4]. These general strategies combined with myriad immunological defenses form a host’s immunological phenotype [4,5]. Qualitative and quantitative differences in aspects of immunological phenotypes have been documented among individuals, populations, and species of animals through space and time [6,7,8,9,10]. As with other plastic phenotypic traits, immunological phenotypes and variation therein are shaped by evolutionary (i.e., genetic) and ecological (i.e., environmental) factors.

Two influential factors are parasite pressure and parasite load [1]. Parasite pressure is largely a characteristic of the broader environment, the specific habitat in which animals live, or both [1]. Exposure to parasite pressure across generations is thought to drive selection and shape immune system evolution, but this has mostly been tested indirectly [1,11]. Parasite load is a trait associated with individual animals [1,12]. The magnitude of parasite load likely depends on parasite pressure and other environmental characteristics, but critically, it also integrates host ecology and immunology [1,11]. Parasite load can be decomposed into several parameters, including the infection status (i.e., presence/absence) and infection intensity. In field studies of wild animals, parasite pressure, parasite load, and host immune defenses are rarely all characterized simultaneously in the same study population.

Knowledge is limited regarding the relative contribution of parasite pressure and parasite load to processes shaping immunological phenotypes in wild animals. Yet understanding these relationships is increasingly important in the light of emerging pathogens that can cause disease in humans [13]. These so-called zoonotic pathogens are often maintained in enzootic cycles by wildlife populations [14], and several zoonotic pathogens have increased in occurrence in recent decades [13]. Of the few studies that have begun to explore ecological immunology in the context of specific pathogens, even fewer study multiple zoonotic pathogens (e.g., [8]). Evaluating relationships among zoonotic pathogens and immune defenses helps in understanding these natural systems and the risks they pose to humans.

We quantified immunological and physiological indices and parasite load parameters in two common species of small rodent from wooded areas in the Netherlands (Appendix A) that were previously shown to differ in their parasite pressure (Appendix A [15]). Moreover, the bank vole (*Myodes glareolus*) and the wood mouse (*Apodemus sylvaticus*), our two study species, differ in their ability to *host* ticks and to *infect* ticks with zoonotic pathogens [6,16]. Bank voles, but not wood mice, acquire resistance to some ectoparasites (e.g., the tick *Ixodes ricinus* [16]), while wood mice mount a stronger antibody-mediated response against zoonotic pathogens than bank voles [6]. Given our interest in both parasite pressure and parasite load and given that infection with one parasite can mediate infection with another (i.e., via mechanisms of co-infection [17]), we took a holistic “parasite assemblage approach”; however, we also maintained a strong focus on vector-borne microparasites. To this end, we screened rodents for an array of ectoparasites, gastrointestinal parasites, and microparasites. (For a full list, see Appendix A.) We also characterized the immunological phenotypes of the same individuals via six indices of the immune system and other allied physiological systems.

We formulated two parasite-related hypotheses. First, if parasite pressure drives immunological phenotype, then populations from different sites are expected to express different immunological phenotypes. In general, higher parasite pressure is thought to select for stronger immune systems [1]. Second, if parasite load drives immunological phenotype, then individuals carrying higher parasite loads are expected to express immunological phenotypes that differ from those carrying lower loads, irrespective of population. The direction of this relationship likely depends on the parasite load parameter under consideration, since some members of the parasite assemblage can be immunostimulatory and others immunosuppressive [17]. Additionally, we expected intrinsic host factors to shape immunological phenotype. In the light of the differences between our study species described above, immunological indices are expected to correlate more strongly with microparasite infection status in wood mice compared to bank voles. Furthermore, immunological indices are expected to correlate positively with body mass (a proxy for age in rodents [18]), a result of immune system development.

## 2. Materials and Methods

### 2.1. Study Sites

Between 13 September and 7 October in 2016, we worked in four 1 ha wooded sites in the Netherlands: Buunderkamp, Herperduin, Maashorst, and Stameren (Appendix A). Details about these sites, including exact locations, have been described previously [15]. We selected these specific sites based on a known gradient in tick burden on rodents (Appendix A [15]) and based on spatial isolation to ensure independent populations of rodents and parasites (the closest neighboring sites, Herperduin and Maashorst, were separated by 5.5 km and a highway).

### 2.2. Rodent Trapping

In each study site, we established a 10 × 10 grid of trapping stations with 10 m between stations. With a pair of Longworth live traps (Heslinga Traps, Groningen, The Netherlands) per station, a grid consisted of 200 traps in total. We activated the traps at 20.00 h on four consecutive evenings and checked them the following mornings at 8.00 h. Traps were closed during the day. We baited the traps with mixed seeds, pieces of carrot and apple, and live mealworms; hay served as insulation. The mealworms were included as food for accidental bycatch of shrews (*Soricidae*). With their utilization of these resources, trapped animals showed no outward signs of stress upon removal from the traps, but any unmeasured effects of capture are expected to have been consistent across animals given the standardized trapping protocol.

Trapped rodents were transferred into a transparent plastic bag to allow for species identification with minimal handling. Once identified, any nontarget species were immediately released. We visually sexed all bank voles and wood mice and then released any females after marking them by fur clipping [19] to facilitate release if recaptured. We weighed all males and evaluated testes development; individuals >20 g or with visible testes were marked and released. By targeting sub-adult males that were not sexually active, we aimed to reduce variation in immunological parameters due to age or reproductive state [2].

### 2.3. Rodent Anesthetization, Sample Collection and Handling

All target males were anaesthetized with a 0.15–0.25 mL injection (depending on body mass) of a 1:2 mixture of xylazine (2 mg/mL) and ketamine (10 mg/mL). After anesthetization, blood was collected via eye extraction, and the animals were euthanized by cervical dislocation. Rodent trapping, anesthetization, euthanization, and all other aspects of the animal experiments were approved in 2016 or earlier by the Central Committee Animal Experimentation in the Netherlands (AVD104002016546), the Animal Welfare Body of Wageningen University (WUR-2016044), and The Netherlands Ministry of Economic Affairs (FF/75A/2013/003).

Blood was collected directly into a tube (MiniCollect^®^ Tube 0.8 mL Z Serum Sep Clot Activator, 450472; Greiner Bio-One B.V., Alphen aan den Rijn, The Netherlands), and samples were kept in a cool box with ice in the field until processing in the lab later the same day. Samples were centrifuged and serum was collected in accordance with the manufacturer’s instructions. After centrifugation, serum was transferred to a fresh microcentrifuge tube and frozen at −20 °C until laboratory analyses, which took place from late October to mid-November 2016.

At the moment of blood collection, we made a blood smear per individual using a drop of the fresh whole blood. Smears were fixed in absolute methanol in the field and stained (Modified Giemsa Stain, GS500-500 ML; Sigma-Aldrich Inc., St. Louis, MO, USA) later in the lab. We also filled one heparinized capillary tube per individual, which was centrifuged at 12,000 rpm for 10 min. From this, we recorded hematocrit values per individual.

From the carcasses, we collected kidneys, liver, spleen, and lungs and some ear tissue in the field for purposes of microparasite screening. The lungs were placed in RNAlater (Thermo Fisher Scientific, Waltham, MA, USA). We kept these samples in a cool box with ice until later the same day, when the lungs were stored at 4 °C and all other organs and tissues and the remaining carcass were stored at −20 °C. Spleen size (mass in mg) was recorded on the day of collection before being frozen. After 3–5 days, we removed the lung samples from the RNAlater and stored them individually in microcentrifuge tubes at −20 °C with the rest of the samples.

### 2.4. Ectoparasite Screening

We screened all carcasses for arthropod ectoparasites. Ectoparasites were collected by combing the fur, checking the skin, and carefully inspecting each carcass’s storage bag. We also collected and stored any ectoparasites that left the carcass during the dissection process. Ectoparasites were stored in 70% ethanol in microcentrifuge tubes (one tube per parasite species per host individual) at −20 °C.

Ectoparasites were counted per species and life stage. Tick and flea species were identified using established identification keys [20,21]. Lice and mite species were identified by Herman J.W.M. Cremers (Utrecht University, Utrecht, The Netherlands), who also confirmed flea species identity.

### 2.5. Gastrointestinal Parasite Screening

Gastrointestinal parasites were isolated following previously described protocols [22]. Briefly, mouse intestines were separated individually into small (duodenum up to ileum) and large intestine (including caecum and colon), and their contents were homogenized in phosphate buffered saline. Parasites were collected by sieving the suspension over a 63 µm mesh size sieve. The stomach was screened for parasites macroscopically and microscopically. Isolated parasites were counted, sexed, identified morphologically, and stored in 70% ethanol.

### 2.6. Screening for Microparasites

DNA and RNA were extracted from tissue samples in a diagnostic laboratory using a robot (MagNA Pure Compact Extraction Robot; Roche, Basel, Switzerland) and Nucleic Acid Isolation Kit I (Roche, Roche, Basel, Switzerland) following the manufacturer’s instructions. To detect potential cross-contamination, we included negative controls in each extraction batch. To minimize contamination and false positives, all main steps (extraction, PCR mix preparation, sample addition, and (q)PCR analyses) were performed in separate air-locked dedicated labs.

We analyzed all nucleic acid extractions with different (multiplex) qPCRs carried out on a LightCycler 480 (Roche Diagnostics Nederland B.V, Almere, The Netherlands). These qPCRs were performed as described previously and based on gene fragments specific for the (vector-borne) microparasites of interest: *Borrelia burgdorferi* s. l. (two targets, [23]), *Borrelia miyamotoi* [24], *Anaplasma phagocytophilum* [25,26], *Candidatus* Neoehrlichia mikurensis [27], *Rickettsia* spp. [28], *Leptospira* spp. [29], and *Bartonella* spp. [30]. We also analyzed the nucleic acid extractions for the presence of four viruses: tick-borne encephalitis virus, Tula hantavirus, Puumala hantavirus, and Eyach virus. For the first three, we performed reverse transcription real-time PCRs as previously described [31,32,33,34]. Positive (plasmid) controls and negative (water) controls were used on every plate.

We used methodology that has not been previously described for two microparasites: *Spiroplasma ixodetes* and Eyach virus. We screened for *S. ixodetes* using exactly the same conditions as for *B. burgdorferi* s. l. but with (rpoBF) 5′-TGTTGGACCAAACGAAGTTG-3′ and (rpoBR) 5′-CCAACAATTGGTGTTTGGGG-3′ as primers and (rpoBP) 5′-(FAM)GCTAACCGTGCTTTAATGGG(BHQ1)-3′ as the probe. For the Eyach virus, we performed a reverse transcription real-time PCR targeting the VP2 of the Eyach virus genome with (EyachF) 5′-TGGCTGACAACATGACGGATA-3′ and (EyachR) 5′-GGCCTCACGATACTTTCGATT-3′ as primers and (EyachP) 5′- ACGGGCTCGGTACTTCGGTTGAGAT-3′ as the probe. We used 20 μL with TaqMan Fast Virus 1-Step Master Mix (Thermo Fisher Scientific, Waltham, MA, USA), 5 μL of sample, and 0.2 μM for the primers and probes for the qPCR, which was performed with a 20 min reverse transcription step at 50 °C, denaturation at 95 °C for 30 s, and 50 cycles of 95 °C for 10 s and 60 °C for 30 s.

Samples positive for *Bartonella* spp. were subjected to conventional PCR and sequencing to determine species identity [35,36]. *Bartonella* sequences were compared with reference sequences from Genbank using the unweighted pair group method with arithmetic mean-based (UPGMA) hierarchical clustering.

### 2.7. Immunological Assays

#### 2.7.1. Haptoglobin

Using a commercially available assay kit (Tridelta PHASE Haptoglobin Assay, TP-801; Tridelta Development Limited, Maynooth, County Kildare, Ireland), we measured circulating concentrations of haptoglobin, which is an acute phase protein that increases in concentration in response to infection, inflammation, or trauma [7,10]. We followed the manufacturer’s instructions, measuring absorbance at 630 nm 5 min after initiation of the color change reaction. We calculated concentrations in our samples using a linear standard curve that we generated from six standards ranging from 0.039 to 1.25 mg Ml^−1^. Most serum samples were analyzed in singlicate, but samples from four individuals were run in duplicate to allow for the quantification of within assay (i.e., within plate) variability. The average coefficient of variation of the four duplicated samples was 5.6%. For each duplicated sample, we used the mean concentration in further analyses.

#### 2.7.2. Hemolysis and Hemagglutination

We measured titers of both complement-driven hemolysis and natural antibody-driven hemagglutination following standard protocols [7,9]. Since the test serum was from mammals, we used blood from a bird (*Columba livia domestica*) to make the required 1% suspension of exogenous erythrocytes. All test samples were analyzed in singlicate, but each of the six assay plates included a duplicate (and in one case, triplicate) sample from a brown rat (*Rattus norvegicus*) to allow for the quantification of within assay (i.e., within plate) and among assay (i.e., among plate) variability. On average, the rat standard exhibited lysis of 4.4 titers and agglutination of 6.1 titers, and the variability (coefficient of variation) was as follows: hemolysis, within 12.4%; hemolysis, among 12.5%; hemagglutination, within 8.3%; and hemagglutination, among 15.5%.

#### 2.7.3. Neutrophil to Lymphocyte Ratio

One author (E.B.) analyzed all of the stained blood smears using a light microscope (at 1000×). Per slide, she counted the neutrophils and lymphocytes until a combined total of up to 100 cells was achieved ($\bar{x}$*_total_* = 83 cells). This process was repeated twice per slide, and the average count of each cell type was used in further analyses of their ratio (neutrophil to lymphocyte ratio).

### 2.8. Data Analysis

We explored relationships between our six response variables related to the immune system and allied physiological systems (haptoglobin concentration (mg mL^−1^), hemolysis (titers), hemagglutination (titers), hematocrit (%), body-mass corrected spleen size (mg g^−1^), and neutrophil to lymphocyte ratio) and several explanatory variables using regression trees [37,38]. We included the following explanatory variables per host individual: site, ecto- and gastrointestinal parasite infestation intensities (i.e., numbers of individuals of each parasite species), microparasite infection statuses (i.e., infected or not with each parasite species, based on molecular techniques), host species identity, and body mass.

Regression trees allowed us to explore the most parsimonious splits in the data using a nonparametric approach that does not assume linearity or independence of data. Furthermore, this analytical approach is ideal for exploring of patterns without a priori hypotheses for specific parasite species. We fitted regression trees in R 3.5.1 [39] using the *rpart* package (v 4.1.13 [40]). Regression trees can be read as a decision tree, where the data are split into two groups at each node. The regression tree for each immunological or physiological response variable was built by recursively partitioning data using an algorithm to split the data into two groups based on the best predictor. This process is repeated for each of the newly formed groups separately, maximizing the deviance in the response variable. A cross-validation step was employed to prune trees using the complexity parameter that accounts for tree complexity and the variance explained. This step resulted in the smallest possible trees with minimum classification error. In contrast to many conventional statistical methods, most regression tree analyses do not calculate statistical significance or *p*-values. To accommodate readers in their interpretation, while adhering to the nonparametric nature of the regression trees, we performed independent 2-group Mann–Whitney U tests [41] to test for differences between groups at each node.

## 3. Results

We sampled 36 rodents (10 bank voles and 26 wood mice) from the four locations (full dataset available in Appendix A). We identified five vector-borne microparasites using qPCR and sequencing: *Bartonella grahamii*, *Bartonella taylorii*, *Borrelia burgdorferi* s.l., *Borrelia miyamotoi*, and *Candidatus* Neoehrlichia mikurensis (Table 1). We found nine ectoparasite species: two tick species (*Ixodes ricinus* and *Ixodes trianguliceps*), four mite species (*Echinonyssus isabellinus*, *Eulaelaps stabularis*, *Haemogamasus nidi*, and *Laelaps agilis*), two flea species (*Ctenophtalmus agyrtes* and *Megabothris turbidus*), and one louse species (*Polyplax serrata*; Table 1). We also found eight gastrointestinal-parasite species: *Aonchoteca murissylvatici*, *Aspiculuris tianjinensis*, *Heligmosomoides polygyrus*, *Heterakis spumosa*, *Hymenolepis diminuta*, *Syphacia petruzewiczi*, *Syphacia stroma*, and an unidentified species of coiled nematode that was not *Trichinella* (Table 1). All serological tests were negative.

We found that simple trees with 3–4 nodes best described the data for all immunological and physiological indices. These trees revealed both differences among sites and differences in relation to parasite infestation or infection. Haptoglobin concentration was highest in animals with an *Ixodes trianguliceps* infestation (Mann–Witney U test, *p* = 0.035) and second highest in animals from Herperduin without *I. trianguliceps* (*p* = 0.022, Figure 1A). Haptoglobin concentration was lowest in animals with a body mass ≥16.5 gram in the other sites (*p* = 0.045, Figure 1A). Hemolysis titer was highest in bank voles (*p* = 0.00093) and intermediate in wood mice with a *Heligmosomoides polygyrus* infection (*p* = 0.050, Figure 1B). Hemagglutination titer was lowest in rodents <15.5 gram (*p* = 0.011) and highest in rodents from three sites (Buunderkamp, Herperduin and Stameren; *p* = 0.033, Figure 1C). Neutrophil to lymphocyte ratio was highest in animals infected with *Borrelia miyamotoi* (*p* = 0.0066) and lowest in animals infested with ≥3 *Laelaps agilis* (*p* = 0.12, Figure 1D). Hematocrit was lowest in rodents <13.5 gram (*p* = 0.0099) and highest in rodents between 13.5 and 17.5 gram (*p* = 0.11, Figure 1E). Body-mass corrected spleen size was highest in animals ≥17.5 gram (*p* = 0.0057) and lowest in animals from Buunderkamp, Herperduin and Maashorst (*p* = 0.036, Figure 1F).

## 4. Discussion

Evaluating relationships between zoonotic pathogens and reservoir hosts, particularly in the context of host immune defenses, can provide new insights into these natural systems and the risks posed to humans. The immunological phenotype of hosts is shaped by both parasite pressure and parasite load [1]. To help to disentangle these two factors, we quantified immunological and related physiological indices and parasite load parameters in bank voles and wood mice from wooded sites in the Netherlands varying in parasite pressure. We found immunological and physiological differences between groups that were and were not infected or infested with parasites, but we also found differences among sites. Defining the biological meaning of such differences, even when using some of the most widely applied assays of ecological immunology, remains challenging, but our results offer some new context to the measured indices. For example, interspecific differences in hemolysis were larger than intraspecific differences related to infestation (parasite load), supporting an earlier idea that this index is relatively invariable in relation to current health status, despite the taxonomic variation it displays [7,9]. In addition, both infestation (parasite load) and site (parasite pressure) associated differences were related to haptoglobin values outside of clinically normal ranges (rodent: 0.25–0.51 mg mL^−1^, murine: 0.00–0.10 mg mL^−1^; both reported by the assay manufacturer). Thus, individuals of both rodent species that were either feeding *I. trianguliceps* ticks or living at Herperduin can be viewed as enduring a type of systemic innate immune response, known as an acute phase response [42]). Overall, we can safely conclude that the measured immunological defenses are shaped by a combination of current parasite load and differences in parasite pressure among spatially distinct sites.

We hypothesized that if parasite pressure drives immunological phenotype, then populations from sites that differ in parasite pressure should express different levels of the measured immune indices. In general, higher parasite pressure is thought to select for stronger immune systems [1]. We found partial support for our hypothesis in the form of differences among sites in terms of haptoglobin, hemagglutination, and body-mass corrected spleen size. In all three cases, one site differed from the other three. This suggests that circumstances specific to some sites might influence parasite pressure (e.g., which parasite species and pathogen strains successfully colonized a site, the overall parasite and pathogen richness or diversity), evolutionary responses to that pressure (e.g., host population genetics), or both. However, the patterns reported here are inconsistent with previously reported differences in parasite pressure, which was characterized by densities of *Ixodes ricinus* and prevalences of several tick-borne pathogens [15]. For example, sites with higher levels of haptoglobin (Herperduin), hemagglutination (Buunderkamp, Stameren and Herperduin), and body-mass corrected spleen size (Stameren) were not the same as the site with the highest parasite pressure in 2013–2014 (Buunderkamp). We did find that endoparasite loads were highest in rodents from Herperduin, which could explain the higher level of haptoglobin at this site, suggesting that a more thorough approach is needed to quantify parasite pressure at a site [2]. Even though sites were on average 30 km apart and were previously shown to have clear differences in tick densities and tick-borne microparasite prevalences [15], these distinctions might not have consistently translated to different immunological phenotypes in measurable ways. Previous studies showing between-site differences in host immunology employed study designs with extreme environmental differences [43].

We also found support for our second hypothesis: individuals burdened with higher parasite loads were expected to differ from those carrying lower loads in terms of the measured immune indices. The direction of effect likely depends on the parasite type (immunostimulatory vs. immunosuppressive) and immune index under consideration. Indeed, we found that the highest and lowest levels of haptoglobin and the neutrophil to lymphocyte ratio were associated with infection or infestation. Concentrations of haptoglobin were higher in individuals infested with the tick *Ixodes trianguliceps*. Haptoglobin is a positive acute phase protein [44], meaning concentrations increase in response to infection, inflammation, or trauma [42], any of which could result from the mouthparts of a tick puncturing the skin of a host and the multiday feeding period that follows [45]. We also found the highest neutrophil to lymphocyte ratios in animals infected with *B. miyamotoi*. This result could be caused by increased neutrophil production or decreased lymphocyte numbers in these infected animals, but neither of these effects were seen in a case-study of a human patient [46]. Furthermore, we found the lowest neutrophil to lymphocyte ratios in animals infested with *Laelaps agilis*. This species of mite is known to be a vector of *Hepatozoon* spp., blood parasites that can infect wild rodents, but for which we did not test our samples [47]. Since *Hepatozoon* spp. can infect leukocytes in rodents [48,49], such an infection might be a mechanism behind the mite infestation effect; however, in other animals (e.g., frogs [50]), neutrophil to lymphocyte ratio was not correlated with infection intensity of *Hepatozoon* spp. Nevertheless, since increased neutrophil to lymphocyte ratios are often associated with stressed or diseased states [51], the lower values we observed in mite infested individuals suggest another process at work (e.g., possible immunosuppression).

Finally, we found differences related to species identity and body mass. Hemolysis was the only immune index for which we found a difference between the two study species: Overall bank voles exhibited higher titers than wood mice. Furthermore and tangentially related to our a priori expectation, hemolysis was the only immune index for which we found an infection-related difference between the two study species. Hemolysis titers were higher in *Heligmosomoides polygyrus* (helminth) infected wood mice but not in similarly infected bank voles. *Heligmosomoides polygyrus*, a commonly used model of helminth infections, is known to regulate immune function in laboratory mice [52]. To our knowledge, our study is the first to show immunological differences associated to natural infection with *H. polygyrus*. It must be noted, however, that the effects of microparasite infection status never showed this type of species dependence. We also found that four immune indices showed three different types of relationships with body mass. Hemagglutination and body-mass corrected spleen size related positively with body mass; haptoglobin related negatively with body mass; and hematocrit showed an optimum at middle body masses. While the effects of age on immune function in rodents are known, if not fully understood [2], our results highlight the complexity of these dynamics. The immune system in wild rodents does not simply “mature” as individuals grow heavier (and older), even if some individual components, such as hemagglutination, show such a pattern here and elsewhere [7,9]. Instead, the observed relationships hint at an influence of body condition or composition, but this potential mechanism could not be investigated given our lack of data on the structural size of individuals [18].

Overall, our findings offer new insights into relationships between specific parasites and immunological and physiological indices, as well as broader differences between species and among sites. Notably, differences related to study site and infection status seemed to exert a greater impact on immune phenotype than host species identity, even though the two rodent study species are thought to play different roles in the maintenance and transmission of tick-borne zoonotic pathogens [6,16]. While our results document influential roles for both the environmental characteristic of parasite pressure and the organismal characteristic of parasite load on the immunological phenotype of wild animals, additional studies linking disease ecology and ecological immunology are needed to better understand the complexities of how host–parasite interactions play out through space and time in different environments. For example, repeated sampling of sites over longer (i.e., multiyear) time periods would be invaluable for characterizing the potential for parasite pressure to drive immune system evolution. Lastly, our study illustrates the possibility and added value of using a holistic approach targeting diverse parasites and multiple aspects of host immunology and physiology when investigating zoonotic pathogens and their vectors and reservoirs [53]. The resulting insights, such as overall differences between species (e.g., hemolysis) and interspecific similarities and differences in the immunomodulatory effects of infection (e.g., neutrophil to lymphocyte ratio and hemolysis, respectively), can help to shape new questions, for example, about host competence [53]. In this study, our focus was on understanding immunological variation, but a similar analytical approach could be used to explore variation in the burden (e.g., infection intensity) of one or more parasites.

## Figures and Tables

**Figure 1 animals-09-01015-f001:**
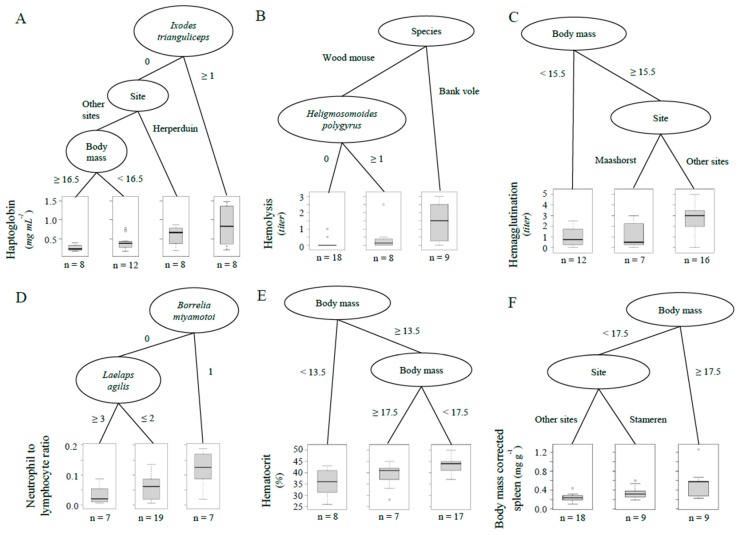
Regression tree showing the most parsimonious split for (**A**) haptoglobin, (**B**) hemolysis, (**C**) hemagglutination, (**D**) neutrophil to lymphocyte ratio, (**E**) hematocrit, and (**F**) body-mass corrected spleen size. Boxplots show the spread of values (median and quantiles) within each group identified by the different splits in the tree. Sample size for each group is given underneath the boxplots. Body mass is given in grams.

**Table 1 animals-09-01015-t001:** Overview of infection with vector-borne microparasites and infestation with ecto- and gastrointestinal parasites in wood mice and bank voles in four Dutch forest sites.

Parasite	Bank Vole (n = 10)		Wood Mouse (n = 26)	
Microparasites				
*Bartonella grahamii* ^1^	0.40		0.04	
*Bartonella taylorii* ^1^	0.10		0.31	
*Borrelia burgdorferi* s.l. ^1^	0		0.04	
*Borrelia miyamotoi* ^1^	0.10		0.23	
*Candidatus* Neoehrlichia mikurensis ^1^	0.30		0.12	
Ectoparasites				
*Ixodes ricinus* (larvae) ^2^	3.20 (0–8)	[0.80]	11.08 (0–80)	[0.92]
*Ixodes ricinus* (nymphs) ^2^	0	[0]	0.08 (0–2)	[0.04]
*Ixodes trianguliceps* (larvae) ^2^	0.70 (0–4)	[0.40]	0.38 (0–4)	[0.15]
*Echinonyssus isabellinus* ^2^	0.10 (0–1)	[0.10]	0	[0]
*Eulaelaps stabularis* ^2^	0	[0]	1.23 (0–27)	[0.19]
*Haemogamasus nidi* ^2^	0	[0]	0.35 (0–9)	[0.04]
*Laelaps agilis*^2^	0	[0]	3.69 (0–27)	[0.58]
*Ctenophtalmus agyrtes* ^2^	0.10 (0–1)	[0.10]	0.08 (0–1)	[0.08]
*Megabothris turbidus* ^2^	0	[0]	0.08 (0–2)	[0.04]
*Polyplax serrata* ^2^	0	[0]	0.85 (0–8)	[0.19]
Gastrointestinal parasites				
*Aonchoteca murissylvatici* ^2^	2.10 (0–14)	[0.30]	5.65 (0–75)	[0.35]
*Aspiculuris tianjinensis* ^2^	15.40 (0–150)	[0.20]	2.31 (0–29)	[0.19]
*Heligmosomoides polygyrus* ^2^	2.80 (0–18)	[0.40]	0.38 (0–3)	[0.31]
*Heterakis spumosa* ^2^	0	[0]	0.04 (0–1)	[0.04]
*Hymenolepis diminuta* ^2^	0.10 (0–1)	[0.10]	0	[0]
*Syphacia petruzewiczi* ^2^	3.80 (0–24)	[0.20]	0	[0]
*Syphacia stroma* ^2^	0	[0]	12.46 (0–76)	[0.58]
Non-*Trichinella* coiled nematode ^2^	0.90 (0–7)	[0.20]	0	[0]

^1^ Prevalence of (vector-borne) microparasites given as ratio of total number of animals with infection. ^2^ Average parasite load as well as prevalence (between square brackets) given as mean and range (in parentheses) of burden and ratio of total number of animals with infestation, respectively.

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
