# Peer review of "Parasite Load and Site-Specific Parasite Pressure as Determinants of Immune Indices in Two Sympatric Rodent Species"

_animals, 2019, doi:10.3390/ani9121015_

Round 1
Reviewer 1 Report
The present manuscript shows a study performed on interesting association between parasite load and parasites pressure related to immune indices
The authors followed the recommended structure with the various section and supplementary data were also included. However, some aspects should be improved, aiming the publication of the manuscript.
Please see below some comments and suggestions, only to further improve the clarity and quality of the manuscript.
Comments:
Line 54-55: please rephrase the sentence. It is very hard to read
Line 55-56: the sentence must be modified.
Line 82: remove the first person in all manuscript (i.e. we, us, our, ect). Please use the impersonal form.
Line 111-116: Probably to understand the zone, the Authors could provide a map of the investigated area
Line. 135-137: Please insert the number of authorization to perform the trapping campaign and the euthanize of specimens.
Line 332: Missing reference
Figure 1-6: This figure must be linked each other in on single figure. The legend is the same in all of this figure. So link the figures and divide it using A,B, C, ecc in the legend
Author Response
Reviewer 1
The present manuscript shows a study performed on interesting association between parasite load and parasites pressure related to immune indices. The authors followed the recommended structure with the various section and supplementary data were also included. However, some aspects should be improved, aiming the publication of the manuscript. Please see below some comments and suggestions, only to further improve the clarity and quality of the manuscript.
RESPONSE: We thank Reviewer 1 for the positive assessment of our manuscript and for the suggestions for improvement. Below we individually address each comment.
Comments:
Line 54-55: please rephrase the sentence. It is very hard to read
RESPONSE: This sentence has been rephrased to improve readability.
The text now reads, “Free-living wild animals are repeatedly exposed to different parasites. These parasites can be found on plants and other animals, in soil and water, and generally throughout the animal’s environment [1,2].”
Furthermore, the entire manuscript has been checked and edited by a native English speaker.
Line 55-56: the sentence must be modified.
RESPONSE: This sentence has been modified.
The text now reads, “When an animal interacts with one or more parasites, that animal’s immune system can respond in different ways [3].”
Line 82: remove the first person in all manuscript (i.e. we, us, our, ect). Please use the impersonal form.
RESPONSE: It is our understanding that Animals and the MDPI family of journals does not have an explicit policy about which person (1st or 3rd) is preferred (or acceptable) in their publications. Rather, this choice depends on the preference of the author(s). In general, over the last decades there has been a movement towards English-language scientific/academic writing in the first person, since this type of writing can increase transparency and readability. For example, Gough (2008) points out that as far back as “1979 the American National Standard for the Preparation of Scientific Papers for Written or Oral Presentation, which represents the views of many scientific organizations, recommends that when a verb concerns action by the author, the first person should be used, especially in matters of experimental design.” Furthermore, colleagues and I have used the first person plural (i.e., "we") in a previous MDPI publication. For these reasons, we prefer to maintain our current use of the 1st person. That said, to help prevent a simplistic and monotonous text, we have tried to avoid overusing sentences written in the first person by also using sentences with other subjects and sentences written in the passive voice.
First-Person Voice
In: The SAGE Encyclopedia of Qualitative Research Methods
By: Noel Gough
Edited by: Lisa M. Given
Book Title: The SAGE Encyclopedia of Qualitative Research Methods
Chapter Title: "First-Person Voice"
Pub. Date: 2012
Access Date: November 12, 2019
Publishing Company: SAGE Publications, Inc.
City: Thousand Oaks
Print ISBN: 9781412941631
Online ISBN: 9781412963909
DOI: https://dx.doi.org/10.4135/9781412963909
Print page: 352
https://methods.sagepub.com/reference/sage-encyc-qualitative-research-methods/n177.xml
Line 111-116: Probably to understand the zone, the Authors could provide a map of the investigated area
RESPONSE: We made a map of the study sites and added it as a supplementary figure.
Line 135-137: Please insert the number of authorization to perform the trapping campaign and the euthanize of specimens.
RESPONSE: In the revision (L139-143), we refer to the three permits (including their numbers) that were required to conduct the research documented in our manuscript.
“Rodent trapping, anesthetization, euthanization, and all other aspects of the animal experiments were approved in 2016 or earlier by the Central Committee Animal Experimentation in the Netherlands (AVD104002016546), the Animal Welfare Body of Wageningen University (WUR-2016044), and the Netherlands Ministry of Economic Affairs (FF/75A/2013/003).”
This information was moved up from L156-159 in our originally submitted manuscript.
Line 332: Missing reference
RESPONSE: The correct reference (i.e., [43]) is present in the text and bibliography, and the extraneous reference has been removed.
Quaye, I.K. Haptoglobin, inflammation and disease. Trans. R. Soc. Trop. Med. Hyg. 2008, 102, 735–742.Figure 1-6: This figure must be linked each other in on single figure. The legend is the same in all of this figure. So link the figures and divide it using A,B, C, ecc in the legend
RESPONSE: We combined figures 1-6 into figure 1A-F as suggested by the reviewer.
Reviewer 2 Report
Ref: animals-635731
Title: Parasite load and site-specific parasite pressure as determinants of immune indices in two sympatric rodent species
Article Type: Type of the Paper (Communication)
The study reports the interaction in wild animals, between parasite pressure, parasite load, and host immune defenses that are rarely all characterized simultaneously in the same study population. The number of animals sampled is not very high, but the amount of analysis performed justifies scientific interest.
The manuscript is well written and interesting.
In addition the study protocol was well performed.
Introduction
Line 93: It would be optimal to specify which microparasites have been searched.
Results
Table 1
It would be important to indicate in the table the average number of isolated parasites (ecto and endo) and the minimum and maximum range, please.
Discussion
Discussions should be rewritten as somewhat confusing to the reader and its are too long.
Interactions with microparasites and ectoparasites should be more specifically specified.
Author Response
Reviewer 2
The study reports the interaction in wild animals, between parasite pressure, parasite load, and host immune defenses that are rarely all characterized simultaneously in the same study population. The number of animals sampled is not very high, but the amount of analysis performed justifies scientific interest. The manuscript is well written and interesting. In addition the study protocol was well performed.
RESPONSE: We also thank Reviewer 2 for the positive assessment of our manuscript and for the suggestions for improvement. We are especially pleased with the acknowledgment that the manuscript is well written and interesting and that the study was performed well. Below we individually address each of Reviewer 2’s comments.
Introduction
Line 93: It would be optimal to specify which microparasites have been searched.
RESPONSE: Since the list of (micro)parasites is rather long, we have chosen to add a reference to the supplementary material in which all parasites that we searched for are listed as columns.
The text now reads, “To this end, we screened rodents for an array of ectoparasites, gastrointestinal parasites, and microparasites. (For a full list, see Supplementary table S3).”
Results
Table 1: It would be important to indicate in the table the average number of isolated parasites (ecto and endo) and the minimum and maximum range, please.
RESPONSE: We have added the range to the already provided averages and separated average parasite load and prevalence more to hopefully clearly show which number represents what.
Discussion
Discussions should be rewritten as somewhat confusing to the reader and its are too long. Interactions with microparasites and ectoparasites should be more specifically specified.
RESPONSE: We struggled to make the most out of this feedback point. While we wish to be as clear as possible for all readers, Reviewer 2 does not offer specifics regarding what about the discussion was found to be confusing. A native English speaker has clarified wording and removed unnecessary text where needed. Furthermore, we have highlighted (in blue text) discussion points related to interactions between rodents and specific microparasites and ectoparasites. We hope that these points serve to satisfy Reviewer 2’s interests in our work without further lengthening the discussion section.
Round 2
Reviewer 1 Report
I not appreciate th personal form, but if journal don't have any guidelines for it, it's ok.